# Geographical variations of food insecurity and its associated factors in Bangladesh: Evidence from pooled data of seven cross-sectional surveys

**Md. Tariqujjaman**[1]*, **Mahfuzur Rahman**[1], **Kinley Wangdi**[2,3], **Gobinda Karmakar**[1], **Tahmeed Ahmed**[1], **Haribondhu Sarma**[2,3]

1 Nutrition and Clinical Services Division, International Centre for Diarrhoeal Disease Research, Dhaka, Bangladesh, 2 Department of Global Health, Research School of Population Health, College of Health & Medicine, Acton, ACT, Australia, 3 The Australian National University, Canberra, Australia

* md.tariqujjaman@icddrb.org

**Data Availability Statement:** Data from the icddr,b Data Repository (icddr,b Datasets) will be provided to interested researchers (Recipients) for purposes

## Abstract

Food insecurity has multiple negative effects on maternal and child health and nutritional outcomes. There is a dearth of up-to-date evidence on the prevalence of food insecurity in Bangladesh based on geographical variations. We investigated the prevalence of food insecurity based on geographical variations and its associated factors. We pooled data from seven cross-sectional surveys conducted in 15,009 households from March 2015 to May 2018. This study was a part of the evaluation of the Maternal Infant Young Child Nutrition Phase 2 programme implemented by BRAC, one of the largest international non-governmental organizations located in Bangladesh that covered rural areas in 26 districts and two urban slums in Dhaka, Bangladesh. We used Household Food Insecurity Access Scale (a widely used scale to measure household food insecurity) to estimate the food insecurity status from the data collected through a face-to-face interview using a structured questionnaire. Hot spot analysis was conducted using the Getis-Ord Gi* statistic. The multiple logistic regression model was applied to explore the associated factors of food insecurity. The food insecurity hotspots were in the northwestern, central-southwestern, and coastal districts of Bangladesh. The overall prevalence of mild, moderate, and severe food insecurity were 12.7%, 13.8%, and 3.5%, respectively. In the adjusted model, household heads and caregivers of children with five or more years of schooling had respectively 42% (adjusted odds ratio (AOR): 0.58, 95% confidence interval (CI): 0.52, 0.64) and 46% (AOR: 0.54; 95% CI: 0.49, 0.61) less likelihood to suffer from food insecurity. Households in the middle (AOR: 0.58, 95% CI: 0.52, 0.65) and rich (AOR: 0.32, 95% CI: 0.28, 0.36) wealth status had lower odds of food insecurity. Food insecurity is widely spread in rural districts of Bangladesh and the degree of vulnerability is higher among the households of the northwestern, central-southwestern, and coastal areas of Bangladesh. Comprehensive interventions including strategies for poverty reduction and education for all might be effective to reduce food insecurity at rural households in Bangladesh.

of secondary data analyses upon approval of a Data Licensing Application & Agreement (Application) by the icddr,b Data Repository Committee. Interested personnel is recommended to consult this with icddr,b's Head of Research Administration Ms. Armana Ahmed (Email: aahmed@icddrb.org).

**Funding:** Children's Investment Fund Foundation had no role in study design, data collection and analysis, decision to publish, or preparation of the manuscript.

**Competing interests:** The authors have declared that no competing interests exist.

## Introduction

Food insecurity is defined as when all members of a given household do not have physical and economic access to sufficient, safe, and nutritious food at all to meet their dietary needs and food preferences for an active and healthy life [1]. People who experience food insecurity are mostly from low- and middle-income countries (LMICs) [2]. Globally, from 2012 to 2014, 805 million people suffered from food insecurity, with limited access to a sufficient quantity of food [3]. In Asia and the Pacific, approximately 351 million people which is about 50% of the global population are predicted to suffer from food insecurity and malnutrition over the next 10 years [4]. Bangladesh is a South Asian country, ranked in 75th position according to the Global Hunger Index out of 107 countries [5]. In the global Food Security Index, among 113 countries, Bangladesh ranked 95th [6]. In Bangladesh, over 15 million people still live in extreme poverty and their daily earnings are less than US$1.90, although Bangladesh has achieved sustainable macroeconomic growth [7].

Food insecurity in the household has multifaceted negative effects on maternal, child, and adolescent health [8, 9]. It is associated with inadequate dietary diversity and nutritional status regardless of age and sex [10]. Micronutrient deficiency, poor diet quality, and poor health status are also associated with food insecurity which impairs the physical, mental, cognitive, and motor development of a child [9, 11, 12]. Food insecurity is also associated with diarrhea, respiratory illness, and stunting [13]. It is also very likely to cause depression, suicidal ideation, and low productivity in schools among adolescents [9]. Given the tremendous negative consequences of food insecurity, we should understand to what extent it does exist in LMICs like Bangladesh and if any certain regions of the country suffer more than other parts of the country, and the factors associated with it.

In Bangladesh, several studies have been conducted regarding food insecurity status. A study explored the factors associated with food insecurity among vulnerable women [14]. Another study was conducted in the seven districts (administrative areas of Bangladesh) of Bangladesh that depicted the association between food insecurity and the dietary diversity and nutritional status of under-five children [10]. A study presented the district-level food insecurity status using per capita calorie intake from the data of the Household Income and Expenditure Survey 2010 and Bangladesh Population and Housing Census 2011 [15]. Although this study explored the district-level variations in the prevalence of food insecurity, it did not manifest the extent of food insecurity and associated correlates of the household's food insecurity in the areas where the large-scale Maternal Infant and Young Child Nutrition (MIYCN) programme has been implemented. Further, the estimates of food insecurity status were not based on recent data. Our study presented food insecurity estimates from recent data between 2015 to 2018. The method of calculating food insecurity in Hossain et al. was different from our study. The calculation of food insecurity based on per capita calorie intake is an indirect or derivative measure [16] whereas the measurement of food insecurity using the Household Food Insecurity Access Scale (HFIAS) is a direct measure, although the latter depends on the recall response of the last 30 days. Both methods have some limitations including indirect mathematical calculation and recall bias. However, the choice of method depends on the resources, context, and questions being answered [16]. In our study, we did not collect calorie intake data which limits us to compare food insecurity using both methods. District-level recent estimates of food insecurity and understanding of their extent are warranted for prioritizing and targeting interventions for specific districts or areas, particularly for resource-limited countries like Bangladesh. The earlier studies were conducted in general settings whereas in our study we intended to understand the food insecurity status, its extent, and correlates in large-scale MIYCN programme implementation areas. Therefore, we anticipate that the results

from this study will provide directives for the programme implementers for prioritizing the programme settings and addressing the factors that correlate with food insecurity in these areas.

## Methods

### Study design and setting

The study utilized pooled data collected at the household level, as part of the evaluation of the MIYCN Phase 2 programme implemented by BRAC (an international development organization). The surveys were conducted from March 2015 to May 2018 in rural areas of 26 districts and two urban slums of Dhaka, Bangladesh. The surveys were conducted in three phases (baseline, midline, and endline) and, to limit the seasonal effect the surveys were conducted at approximately the same period of the year. The MIYCN programme used concurrent evaluation which is an innovative approach to evaluating complex real-world programmes. The evaluation method including survey timelines, evaluation activities, evidence, course correction, and study procedure has been discussed elsewhere [17, 18].

### Sample size

We considered a 50% prevalence of micronutrient powder (MNP) coverage, a precision of ±10%, and a design effect of 2 for calculating the sample size. Our estimated minimum sample size was 192 households per district for caregivers of 6-59-month-old children. The detailed sample size calculation was presented in other papers of this project [19, 20]. For this article, we pooled data from seven surveys since the first survey of MIYCN did not include all the food insecurity indicators. In this study, we included 15,009 respondents as our study participants.

### Sampling procedure

A multi-stage sampling technique was applied to select the households. First, a systematic random sampling technique was used to select communities or primary sampling units (PSUs). To obtain the minimum estimated sample size, 16 PSUs were selected from a list of targeted PSUs, which were sorted by district and sub-districts within the districts. In the second stage of sampling, a physical map-segment sample approach was exercised to segment the selected communities or PSUs. Finally, the Expanded Programme on Immunization (EPI-5) method was applied by spinning a bottle/pen placed in the center of the segment, counting the households along that route, and picking the fifth household. We used this EPI-5 method to select households because we intended to see the coverage of a large evaluation programme. An android-based tablet (smartphone device) developed in the Open Data Kit platform was used for data collection.

### Data collection

We interviewed the caregivers of children aged 6–59 months. The interviewer read the aims, and process of the study before the interview in the language interviewee understood. They were allowed to ask questions or raised any concerns about the study. After they were satisfied, the interviewer took the well-informed written consent from the interviewee. We used a structured questionnaire to collect the data. The questionnaire contains socio-demographic, health, and nutrition-related standard modules for caregivers, children, and households. The interviews were administered by trained interviewers and lasted 40–60 minutes. To check the quality of data, the field research supervisor re-interviewed 6–10% of the original interviews.

## Outcome measure

The outcome variable in this study was food insecurity status (categorized as food secure, mild, moderate, and severe food insecure). These four categories were further merged into two categories: food secure and food insecure (adding mild, moderate, and severe food insecure into the food insecure category) for the regression analysis. Information on food insecurity status was obtained using the HFIAS developed by Food and Nutrition Technical Assistance [21]. We assessed the food insecurity status based on the last 30 days' recall responses from 9 questions (**S1 Table**). The questions were asked on how often the level of concern, lack of access to, variety, and quantity of food happened. The response to each question was between 0 to 30. We made scoring based on the responses as 0 = 0, 1–2 = 1, 3–10 = 2, and 11–30 = 3. Thus, the score ranged from 0 to 27 for 9 questions. We then categorized the scores as 0–1 = food secure, score 2/7 = mild food insecure, 8/14 = moderate food insecure, and score 15–27 = severe food insecure [22].

## Covariates measure

The covariates were identified from the review of relevant literature and included the known ones that might have a potential association with food insecurity. The covariates were household size (defined as the number of members living in each household and categorized as: <5 and ≥5 members), child's sex (male and female), households with the number of children aged 6–59 months (categorized as one and two or more), caregivers age (categorized as <25 years and ≥25 years), caregiver's education (categorized as <5 years of schooling and ≥5 years of schooling), father's age (categorized as <30 years and ≥30 years), father's education (categorized as <5 years of schooling and ≥5 years of schooling), and wealth index (categorized as poor, middle, and rich). The household size, caregiver's age, and father's age were categorized based on the median value (one category for values below the median and other values at and above the median). The wealth index was calculated based on household materials (e.g. materials used for the floor, roof, and wall of the house), and household assets (including the type of latrine used and sources of drinking water) by performing a principal component analysis [23]. We also included the district as an independent variable in the regression model to explore the likelihood of food insecurity in other districts compared to the lowest food-insecure district (i.e. Chuadanga). Additionally, we included the survey year as a variable (2015, 2016, 2017, and 2018) in the multiple regression model to control the time variations in food insecurity.

## Data analysis

All statistical analyses were performed using the statistical software package Stata 15.1 (Stata Corporation, College Station, TX, USA). Data analyses included descriptions of the study population and estimates of food insecurity statuses. We presented these estimates with frequencies and percentages. Univariate and multivariable logistic regression models were performed to determine the statistical associations between food insecurity and sociodemographic or other covariates. The results are presented as odds ratios (ORs) and adjusted odds ratios (AORs) with respective 95% confidence intervals (CIs). The covariates that were significantly associated with food insecurity in the simple regression models as well as other covariates that were not statistically significant but conceptually linked with household food insecurity (for example, household size, household head's age), were included in the final multivariable regression model.

The general equation of the multivariable logistic regression model is:

$$y = \log\left[\frac{p}{1-p}\right] = \alpha_0 + \beta_1 x_1 + \beta_1 x_2 + \beta_1 x_2 + \ldots + \beta_n x_n$$

Where:

y = Outcome variable (food insecurity status)

$p$ = probability of households being food insecure

$\alpha_0$ = Intercept (constant)

$\beta_1$, $\beta_2$, $\beta_3$, . . . . . . $\beta_n$ = Coefficients of the respective independent variables

$x_1$, $x_2$, $x_3$, . . . $x_n$ = Independent variables (Household size, household head's age, education of household head. . .survey year, etc.)

Cluster-adjusted and weighted analyses were performed by using the 'svy' command in Stata. We checked the collinearity among the covariates and found the mean-variance inflation factor 1.15, indicating negligible collinearity among the covariates. A p-value of <0.05 was considered statistically significant.

Household location was the unit of spatial analysis. An electronic map of district boundaries in shapefile format was obtained from the DIVA-GIS database (https://www.diva-gis.org/). Food insecurity status was mapped in the located households to provide distribution across the different districts. Hot spot analysis was conducted using the Getis-Ord Gi* statistic. The Getis-Ord Gi* statistics is better suited for our dataset because in our data both high and low values are clustered spatially [24]. Gi* statistics works by looking at each feature within the context (in our study all the study households) of neighboring features. A feature with a high value is interesting but may not be a statistically significant hot spot. To be a statistically significant hot spot, a feature will have a high value and be surrounded by other features with high values as well. The Gi* statistic is a z-score that identifies areas of higher or lower values by comparing them to a normal probability distribution and provides a measure of the local concentration of positive results. Each household location was assigned a value of "1" if the households fell in the insecure category or "0" if secured. For the conceptualization of spatial relationships, we used the Fixed Distance Band; this statistic compares the spatial dependency of food insecurity between the households to identify hot spots and cold spots. A high z-score and a small p-value for a feature indicate a spatial clustering of high values. A low negative z-score and a small p-value indicate a spatial clustering of low values (cold spot). The higher (or lower) the z-score, the more intense the clustering (hot spot) [25]. The Getis-Ord Gi* statistic was used to classify households into hot spots and cold spots with 90%, 95%, and 99% confidence. For spatial analyses, 14,961 unique household locations (longitudes and latitudes) were used while 48 households were excluded due to the wrong locations (such as the longitudes and latitudes were located in the sea or outside the country). Mapping and hotspot analysis was done in ArcMap version 10.7.1 (ESRI, Redlands, CA).

### Ethical clearance

The study protocol was reviewed and approved by the Institutional Review Board of icddr,b which consisted of two committees: the Research Review Committee and the Ethics Review Committee.

## Results

### Sociodemographic characteristics

Of the households surveyed, 59% of them were with five or more members living together. Seventy-three percent of the household heads were of 30 years or above. The median age of caregivers was 25 years. More than 77% of household heads and caregivers completed five or more years of schooling. Around 15% of the household had at least two children of 6–59 months and about 18% of the households reported the birth of children within the last year. The distribution of the wealth index was in similar patterns (33.0%) across three categories including poor, middle, and rich (**Table 1**).

**Table 1. Socio-demographic characteristics of the study population (n = 15,009).**

| Variables | Percentage | Number |
|---|---|---|
| **Household size** | | |
| ≥5 members | 58.73 | 8843 |
| **Household head's age** | | |
| ≥30 years | 73.28 | 11,036 |
| **Education of household head** | | |
| ≥5 years of schooling | 64.39 | 9796 |
| **Education of caregivers** | | |
| ≥5 years of schooling | 77.52 | 11,629 |
| **Child's sex** | | |
| Female | 47.65 | 7204 |
| **Number of children aged 6–59 months** | | |
| Two or more | 14.51 | 2144 |
| **Most recent birth** | | |
| ≤12 months | 17.52 | 2617 |
| **Wealth index** | | |
| Poor | 34.38 | 5147 |
| Middle | 32.51 | 4888 |
| Rich | 33.11 | 4974 |
| **Survey year** | | |
| 2015 | 35.01 | 4811 |
| 2016 | 37.91 | 4961 |
| 2017 | 8.40 | 1540 |
| 2018 | 18.67 | 3697 |

Percentages with numbers are presented for each category. Variables with only two categories (example household size) the percentage and number are presented only for one category.

## Distribution of food insecurity

The prevalence of mild, moderate, and severe food insecurity was 12.7%, 13.8%, and 3.5%, respectively (**Fig 1**). The mild food insecurity status declined from 18% in 2015 to 6% in 2018. Moderate and severe food insecurity declined from 17% and 5% in 2015 to 11% and 2% in 2016 but increased in 2017 (19% and 5%) and then declined to 11% and 3% in 2018 (**S1 Fig**). The vulnerable districts with severe food insecurity status were Nilphamari (18.6%) followed by Lalmoinirhat (14.1%), Rangpur (8.3%), Kurigram (7.4%), and Barguna (5.9%). The moderate food-insecure districts were Lalmonirhat (30.6%), Nilphamari (26.4%), Kurigram (25.1%), and Gaibandha (24.1%). Further, in the case of mild food insecurity, the most vulnerable districts were Kurigram (24.1%), Mymensingh (21.1%), Manikgonj (18.7%), and Cox's Bazar (18.3%) (**Fig 2**).

## Geographical variations of food insecurity

Among 26 districts, the most vulnerable districts in terms of food insecurity were the North and South (coastal) territorial regions (**Fig 3**). The distribution of food insecurity by households is shown in **Fig 4**. Food insecurity was reported in 4,354 households while 10,607 households had food security. Hot spots of food insecurity were located in the northwestern, central-southwestern, and coastal parts of the country including Nilphamari, Kurigram, and Lamonirhat districts. While cold spots were located in Bogra, Meherpur, Norsingdi, and Feni (**Fig 5**).

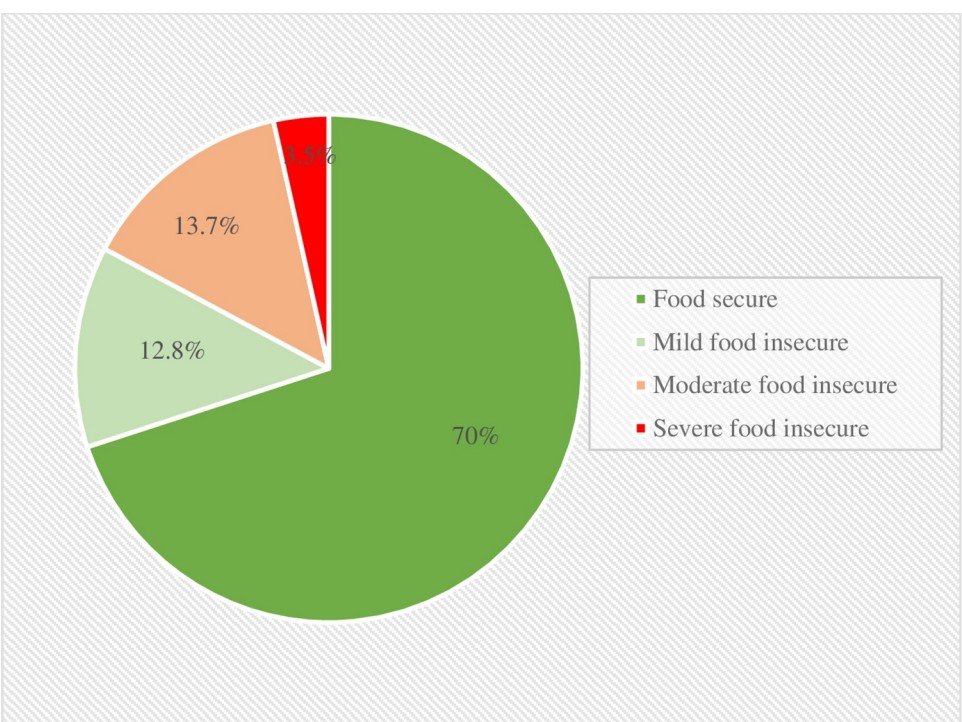

**Fig 1. Prevalence of food insecurity status in 26 districts of Bangladesh.**

### Associated factors of food insecurity

In the univariate models, the education of household heads, caregiver's age, education of caregivers, wealth index, district, and survey year were significantly associated with food insecurity. After adjusting the covariates in the multivariable model, the household heads and caregivers of the children with five or more years of schooling had respectively, 42% (AOR: 0.58, 95% CI: 0.52, 0.64), and 46% (AOR: 0.54, 95% CI: 0.49, 0.61) less likely to suffer from food insecurity than the households with the household head and caregiver having no schooling or below five years of schooling. The households of the middle (AOR: 0.58, 95% CI: 0.52, 0.65), and rich (AOR: 0.32, (95% CI: 0.28, 0.36) wealth status had lower odds to be food insecure than the poor households. Compared to the survey year 2015, we found lower odds of food insecurity in the following survey years indicating the decline of the food insecurity status over time (**Table 2**).

### Discussion

We found one out of three rural households in Bangladesh suffered from food insecurity which corresponds with the findings of another study [30]. Food insecurity was higher in the northwestern, followed by central-southwestern and coastal districts of Bangladesh compared to the other districts. Poverty is the primary cause of food insecurity in rural regions [26] and we also found the wealth status of the households as an associated factor of food insecurity. In addition, the education of household heads and the education of mothers or caregivers of the children also came up as potential factors associated with food insecurity.

The hot spot of food-insecure households was located in the northwestern (Lalmonirhat, Nilphamari, Kurigram, Rangpur, Gaibandha, Mymensingh, and Sylhet), followed by central-southwestern (Rajbari, Faridpur, Magura, Jessore, and Madaripur) and coastal districts (Cox's

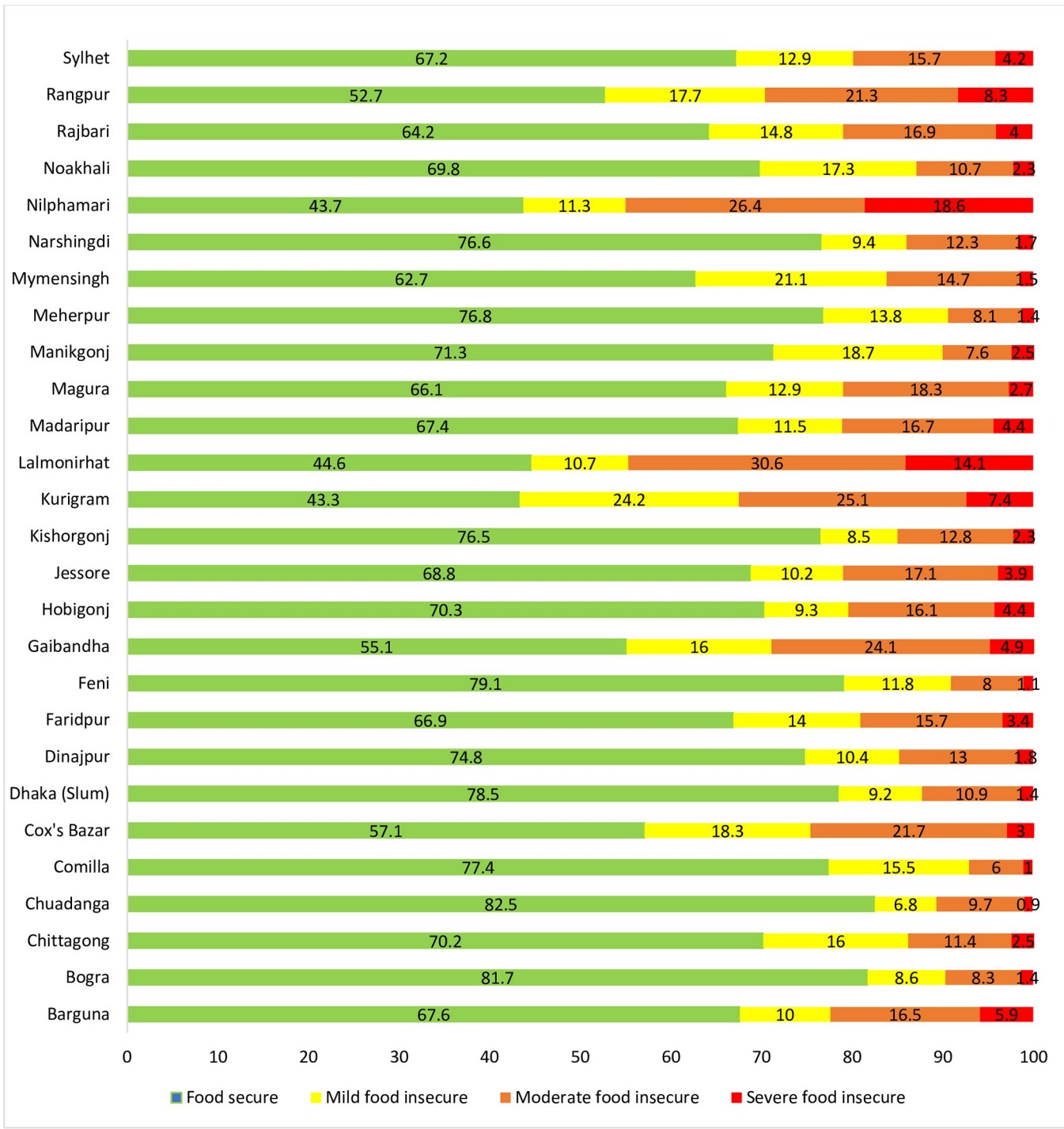

**Fig 2. District-wise prevalence of food security and different categories of food insecurity status.**

Bazar, and Barguna) of Bangladesh. Most of the people in these regions have low income (Lalmonirhat, Nilphamari, Kurigram, Rangpur, Gaibandha, and Mymensingh) and the literacy rate is also low (Sylhet) [27, 28]. A major portion of people living in the districts of the northern parts is suffering from a cyclical phenomenon of poverty and hunger called "Monga" [29].

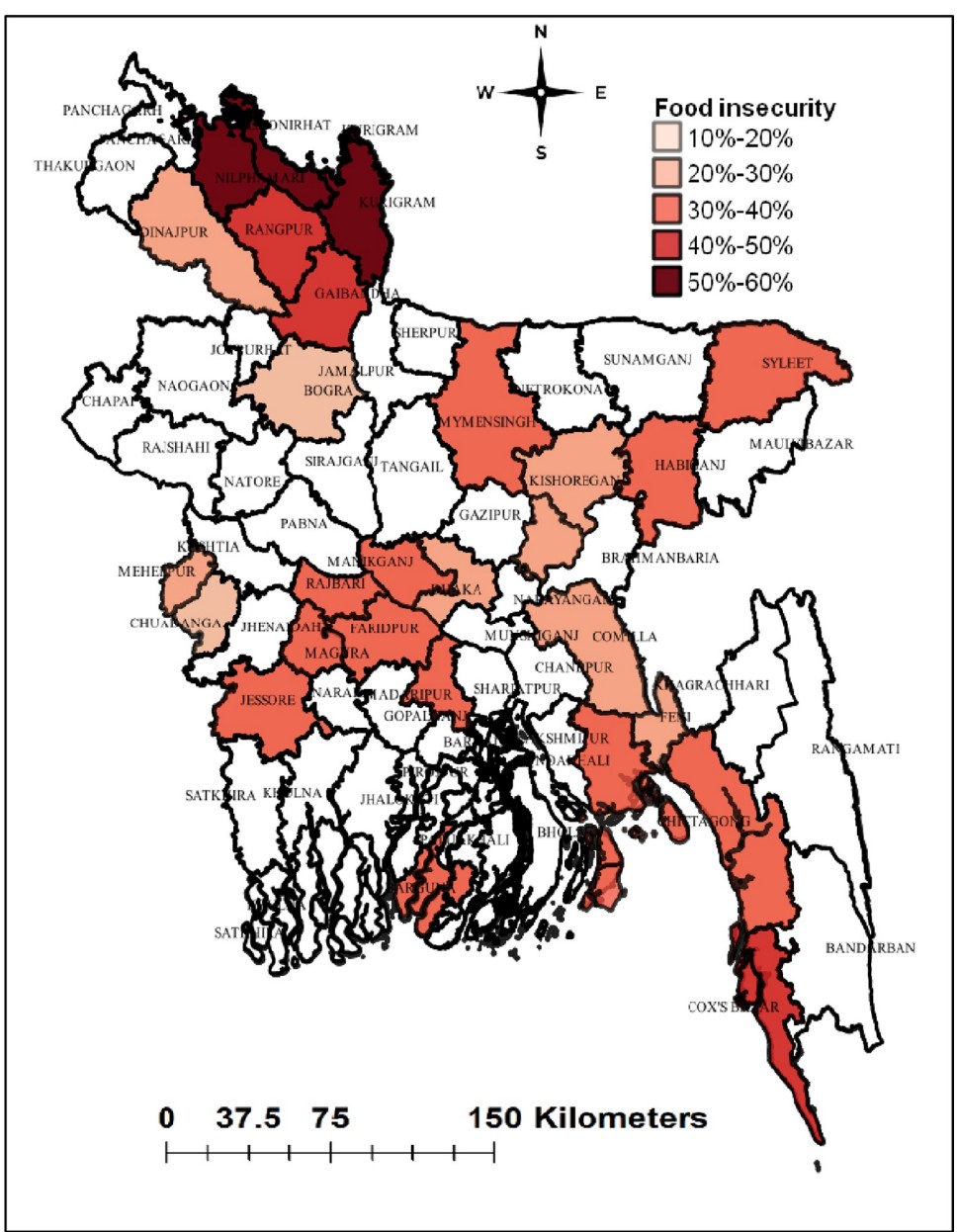

**Fig 3. Geographical variations in the prevalence of food insecurity status by districts.**

Poor individuals are unable to buy food and have limited access to food, forcing them to be food insecure [30]. In contrast, higher household income increases purchasing power and thus increases the household's accessibility to food [31]. Therefore, interventions on poverty alleviation including cash and food transfer programme in the northwestern, central-southwestern, and coastal districts might be effective to reduce the food insecurity [32]. Moreover, there is a link between poverty and household head's education, so proper monitoring and strengthening of the school-based food for education programme might be effective to increase the literacy rate among the people of vulnerable areas of Bangladesh [33].

People living in coastal districts (Cox's Bazar, Barguna) often experience the effects of cyclones. In addition, these areas are subjected to significant natural calamities including

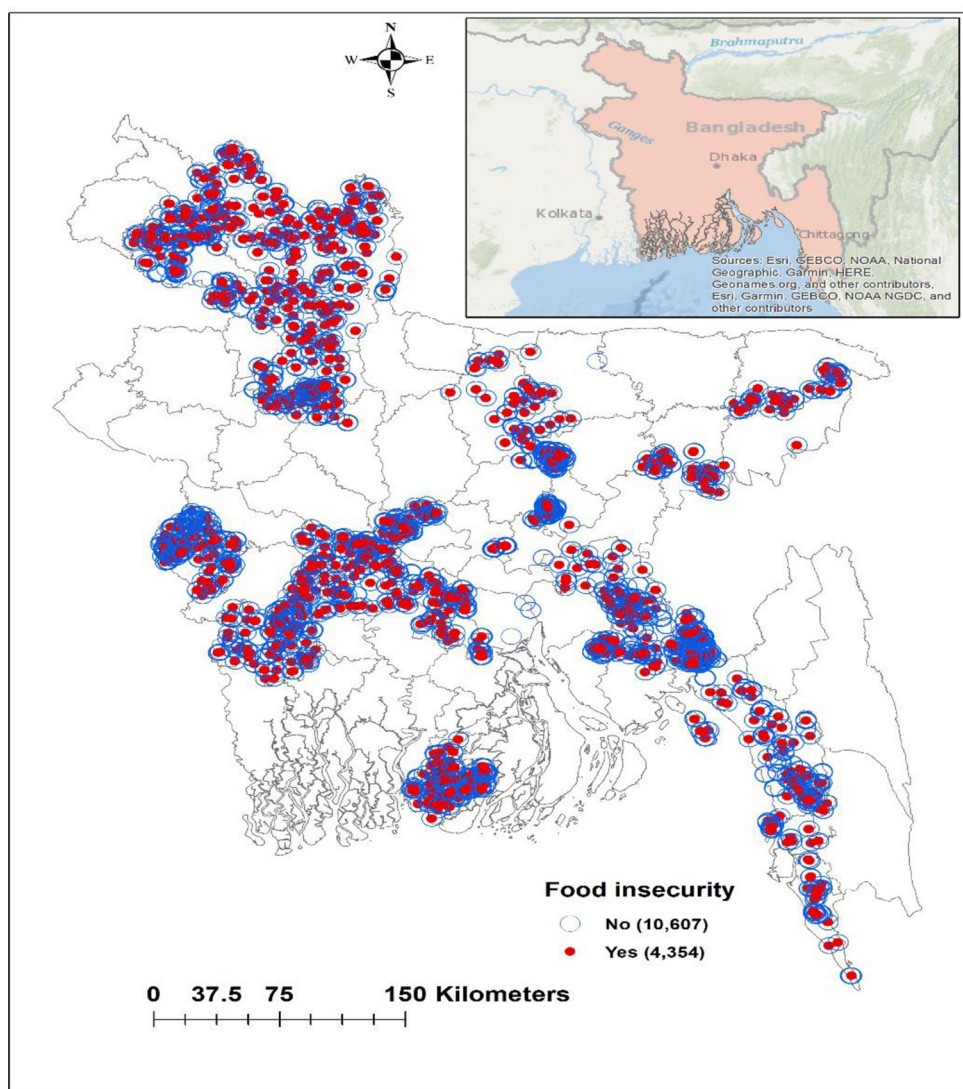

**Fig 4. Distribution of food insecurity across the country by survey locations.**

floods, droughts, river bank erosion, and other natural disasters [34]. These natural calamities contribute to acute food insecurity through reduced agricultural production and work opportunities for income generation. Most of the people in the coastal areas depend on fishing and farming agricultural lands for their livelihood. Frequent natural calamities and salinization of soil reduce crop production and pose long-term environmental degradation leading to food insecurity in these areas [35]. Following northwestern and coastal districts, the food insecurity hotspot is also located in the central-southwestern districts (Rajbari, Faridpur, Madaripur, Magura, and Jessore). In these areas, floods, river bank erosions droughts are the consequences of climate change which might have effects on high food insecurity. Therefore, relevant policy-makers should take initiatives for motivating farmers to produce climate-resilient crops in order to address the problem of food insecurity in these vulnerable districts. Different nutrition-sensitive and specific interventions should be reinforced in these regions since food security is directly linked to childhood malnutrition [36]. Apart from those different interventions, programmes for adolescents, pregnant women, and lactating women should be proactively

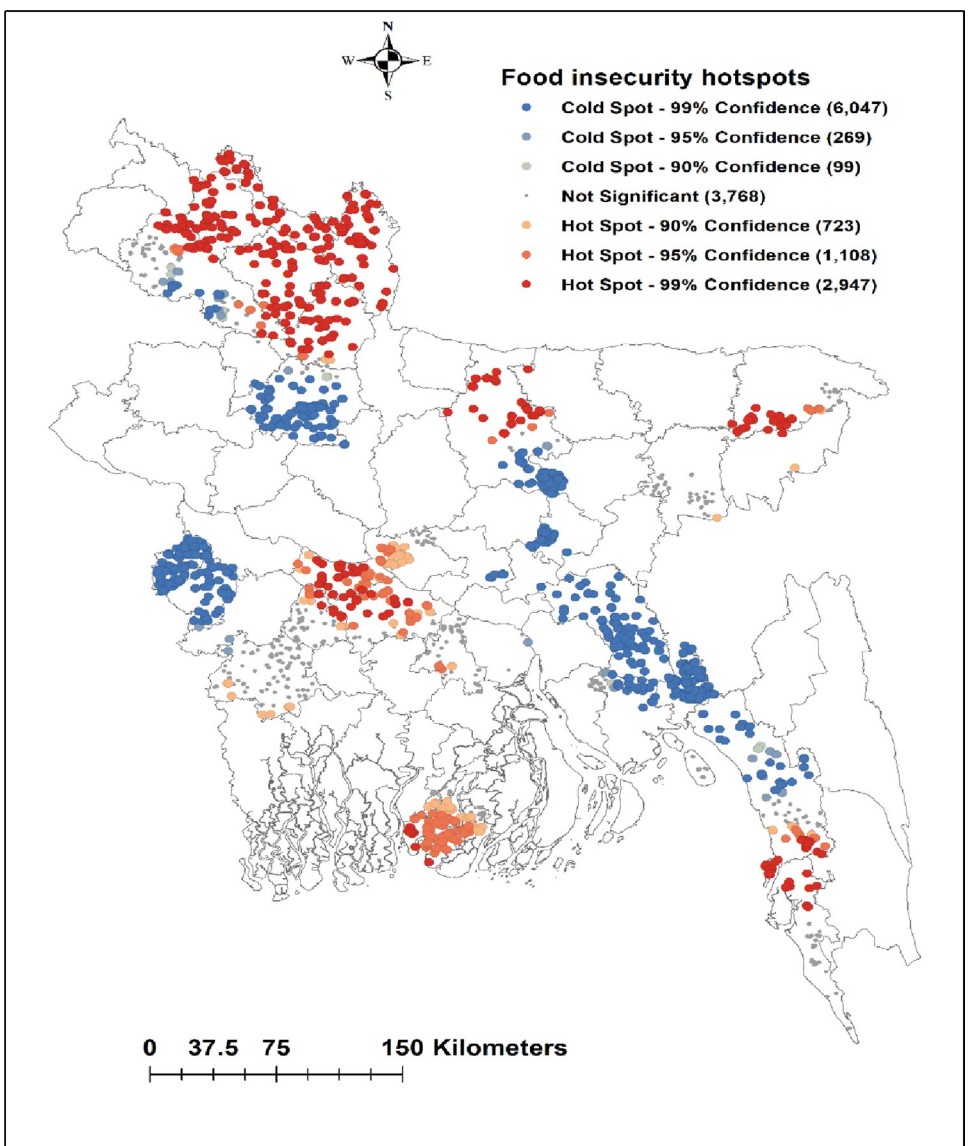

**Fig 5. Hot spot analysis of food insecurity using Getis-Ord Gi* statistics.**

implemented and strengthened to reduce maternal malnutrition because they might be more vulnerable under the conditions of food insecurity in the households [37–39].

Several studies have demonstrated the negative consequences of food insecurity on maternal and child health [9, 11, 12]. However, the situation of food insecurity may be worsened if the households experience unprecedented events such as lockdowns due to COVID-19. During the COVID-19 lockdown period, in Bangladesh, 90.0% of households suffered from different grades of food insecurity [40]. Another Bangladeshi study found that before the COVID-19 pandemic, 5.6% and 2.7% of households experienced moderate and severe food insecurity, and, moderate and severe food insecurity increased to 37% and 15% respectively, during the lockdown [41]. This also had significant mental health effects mostly amongst low-income people due to the loss of income during the lockdown [42]. The prevalence of food security status in these two studies was higher than in our study since these studies were conducted during

**Table 2. Unadjusted and adjusted associated factors of food insecurity status in Bangladesh.**

| Variables | OR (95% CI) | p-value | ¶AOR (95% CI) | p-value |
|---|---|---|---|---|
| **Household size** | | | | |
| <5 members | Ref. | | Ref. | |
| ≥5 members | 0.94 (0.87, 1.02) | 0.143 | 0.95 (0.87, 1.05) | 0.335 |
| **Household head's age** | | | | |
| <30 years | Ref. | | Ref. | |
| ≥30 years | 0.96 (0.87, 1.04) | 0.314 | 0.82 (0.72, 0.92) | 0.001 |
| **Education of household head** | | | | |
| <5 years of schooling | Ref. | | Ref. | |
| ≥5 years of schooling | 0.37 (0.34, 0.41) | <0.001 | 0.58 (0.52, 0.64) | <0.001 |
| **Caregiver's age** | | | | |
| <25 years | Ref. | | Ref. | |
| ≥25 years | 1.38 (1.27, 1.50) | <0.001 | 1.31 (1.16, 1.47) | <0.001 |
| **Education of caregivers** | | | | |
| <5 years of schooling | Ref. | | Ref. | |
| ≥5 years of schooling | 0.34 (0.31, 0.38) | <0.001 | 0.54 (0.49, 0.61) | <0.001 |
| **Child's sex** | | | | |
| Male | Ref. | | Ref. | |
| Female | 0.96 (0.89, 1.04) | 0.337 | 0.95 (0.87, 1.04) | 0.256 |
| **Number of children aged 6–59 months** | | | | |
| One | Ref. | | Ref. | |
| Two or more | 1.09 (0.97, 1.21) | 0.144 | 1.16 (1.02, 1.31) | 0.019 |
| **Most recent birth of the household** | | | | |
| >12 months | Ref. | | Ref. | |
| ≤12 months | 1.00 (0.89, 1.11) | 0.999 | 1.02 (0.91, 1.15) | 0.706 |
| **Wealth index** | | | | |
| Poor | Ref. | | Ref. | |
| Middle | 0.57 (0.51, 0.63) | <0.001 | 0.58 (0.52, 0.65) | <0.001 |
| Rich | 0.28 (0.25, 0.32) | <0.001 | 0.32 (0.28, 0.36) | <0.001 |
| **District** | | | | |
| Chuadanga | Ref. | | Ref. | |
| Barguna | 2.01 (1.47, 2.75) | <0.001 | 2.65 (1.88, 3.75) | <0.001 |
| Bogra | 0.94 (0.68, 1.29) | 0.697 | 1.58 (1.12, 2.22) | 0.008 |
| Chittagong | 1.79 (1.29, 2.48) | 0.001 | 2.85 (2.02, 4.03) | <0.001 |
| Comilla | 1.22 (0.90, 1.67) | 0.197 | 2.14 (1.54, 2.98) | <0.001 |
| Cox's Bazar | 3.16 (2.22, 4.51) | <0.001 | 4.95 (3.34, 7.35) | <0.001 |
| Dhaka | 1.42 (1.03, 1.95) | 0.033 | 2.34 (1.67, 3.28) | <0.001 |
| Dinajpur | 2.08 (1.52, 2.85) | <0.001 | 2.83 (1.93, 4.14) | <0.001 |
| Faridpur | 1.11 (0.77, 1.60) | 0.574 | 1.76 (1.22, 2.53) | 0.003 |
| Feni | 3.43 (2.38, 4.94) | <0.001 | 3.78 (2.47, 5.80) | <0.001 |
| Gaibandha | 1.78 (1.18, 2.68) | 0.006 | 2.08 (1.42, 3.05) | <0.001 |
| Hobigonj | 1.91 (1.41, 2.57) | <0.001 | 2.81 (2.04, 3.88) | <0.001 |
| Jessore | 1.29 (0.91, 1.82) | 0.148 | 1.57 (1.11, 2.23) | 0.011 |
| Kishorgonj | 5.50 (3.77, 8.03) | <0.001 | 5.73 (3.72, 8.84) | <0.001 |
| Kurigram | 5.22 (3.79, 7.18) | <0.001 | 6.04 (4.19, 8.71) | <0.001 |
| Lalmonirhat | 2.03 (1.41, 2.93) | <0.001 | 2.80 (1.72, 4.55) | <0.001 |
| Madaripur | 2.15 (1.55, 2.99) | <0.001 | 2.73 (1.82, 4.11) | <0.001 |
| Magura | 1.69 (1.24, 2.31) | 0.001 | 2.31 (1.67, 3.20) | <0.001 |

*(Continued)*

**Table 2.** (Continued)

| Variables | OR (95% CI) | p-value | ¶AOR (95% CI) | p-value |
|---|---|---|---|---|
| Manikgonj | 1.27 (0.88, 1.84) | 0.202 | 1.88 (1.22, 2.88) | 0.004 |
| Meherpur | 2.50 (1.75, 3.56) | <0.001 | 3.01 (1.98, 4.57) | <0.001 |
| Mymensingh | 1.28 (0.89, 1.84) | 0.179 | 2.01 (1.39, 2.91) | <0.001 |
| Narshingdi | 5.42 (3.59, 8.16) | <0.001 | 6.09 (4.01, 9.26) | <0.001 |
| Nilphamari | 1.82 (1.34, 2.49) | <0.001 | 3.12 (2.20, 4.41) | <0.001 |
| Noakhali | 2.34 (1.69, 3.25) | <0.001 | 2.80 (1.88, 4.15) | <0.001 |
| Rajbari | 3.77 (2.47, 5.74) | <0.001 | 4.71 (2.88, 7.71) | <0.001 |
| Rangpur | 2.05 (1.41, 2.99) | <0.001 | 3.01 (2.06, 4.40) | <0.001 |
| Sylhet | 2.01 (1.47, 2.75) | <0.001 | 2.65 (1.88, 3.75) | <0.001 |
| **Survey year** | | | | |
| 2015 | Ref. | | Ref. | |
| 2016 | 0.48 (0.42, 0.55) | <0.001 | 0.58 (0.49, 0.68) | <0.001 |
| 2017 | 0.73 (0.62, 0.85) | <0.001 | 0.48 (0.40, 0.59) | <0.001 |
| 2018 | 0.34 (0.30, 0.39) | <0.001 | 0.40 (0.34, 0.46) | <0.001 |

Outcome variable: Food insecurity status (two categorizations (food security and food insecurity) based on household food insecurity access scale. OR- odds ratio; AOR-adjusted odds ratio; CI- confidence interval; Ref- reference; ¶Multiple logistic regression model adjusted for household size, household head's age, education of household heads, caregiver's age, education of caregivers, child's sex, number of children aged 6–59 months, most recent birth of household, wealth index, district and survey year.

the COVID-19 period when anxiety regarding food insecurity was higher among the people. Therefore, there should have readiness for the government to tackle the acute food insecurity stemming from such an unprecedented event.

## Strengths and limitations of the study

This study has several strengths. The latest estimate of food insecurity provides insight into the current scenario at the national level with the representative sampled districts across the country. Fine resolution at the district level estimates will help the government, policymakers, planners, and other non-governmental organizations to allocate resources and also implement different intervention programmes targeting the most vulnerable districts rather than a blanket resource allocation. The large sample size and random sampling technique of this study from different parts of Bangladesh make the estimate robust and representative. This study has limitations too. Data were collected mainly from the rural part of Bangladesh where a large-scale MIYCN program was implemented and so, the results might not be representative of the entire Bangladesh where the MIYCN programme are not functioning. Data were collected from two urban slums in Dhaka which do not necessarily represent the food insecurity status of urban areas in Bangladesh. To address these problems, further research is warranted to estimate urban food insecurity status. Furthermore, due to the cross-sectional nature and limited survey years, we cannot make any causal inferences as well as establish the temporality. We suggest a longitudinal study with a wider time frame to explore the cause-effect relationship and temporal trends of food insecurity as well as to monitor the food insecurity status and accordingly allocate the resources in the vulnerable districts in rural Bangladesh.

## Conclusion

Food insecurity is widely spread in rural districts of Bangladesh and the degree of vulnerability is higher among the households of the northwestern, central-southwestern, and coastal areas

of Bangladesh. The government should focus on target-oriented interventions to reduce food insecurity in those particular areas. Comprehensive intervention including strategies for poverty reduction and education for all might be effective to lower the food insecurity status in rural Bangladesh.

## Supporting information

**S1 Fig. Year-wise prevalence of food insecurity status of Bangladesh.**
(DOCX)

**S1 Table. Household hunger scale questionnaire.**
(DOCX)

## Acknowledgments

We are thankful to all the caregivers and children for participating in this study, the data collectors for collecting the data, the field supervisor for their tireless support, Zillur Rahman for administrative support, and Mohammad Ashraful Islam for the managerial role. We would also like to acknowledge GAIN for technical support and BRAC for implementing the programme. icddr,b is grateful to the Governments of Bangladesh, Canada, Sweden, and the United Kingdom for providing core/unrestricted support.

## Author Contributions

**Conceptualization:** Md. Tariqujjaman, Haribondhu Sarma.

**Data curation:** Md. Tariqujjaman.

**Formal analysis:** Md. Tariqujjaman, Kinley Wangdi, Gobinda Karmakar.

**Funding acquisition:** Tahmeed Ahmed, Haribondhu Sarma.

**Investigation:** Mahfuzur Rahman, Tahmeed Ahmed, Haribondhu Sarma.

**Methodology:** Md. Tariqujjaman, Mahfuzur Rahman, Gobinda Karmakar, Tahmeed Ahmed, Haribondhu Sarma.

**Project administration:** Mahfuzur Rahman, Haribondhu Sarma.

**Software:** Md. Tariqujjaman, Kinley Wangdi.

**Supervision:** Mahfuzur Rahman, Tahmeed Ahmed, Haribondhu Sarma.

**Validation:** Mahfuzur Rahman, Kinley Wangdi, Haribondhu Sarma.

**Writing – original draft:** Md. Tariqujjaman, Mahfuzur Rahman, Kinley Wangdi, Gobinda Karmakar, Tahmeed Ahmed, Haribondhu Sarma.

**Writing – review & editing:** Md. Tariqujjaman, Mahfuzur Rahman, Kinley Wangdi, Gobinda Karmakar, Tahmeed Ahmed, Haribondhu Sarma.

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
