## [Decision Letter · Decision Letter 0]

11 Apr 2022

PONE-D-21-34476Geographical Variations of Food Insecurity and its Associated Factors in Bangladesh: Evidence from Pooled Data of Seven Cross-sectional SurveysPLOS ONE

Dear Dr. Tariqujjaman,

Thank you for submitting your manuscript to PLOS ONE. After careful consideration, we feel that it has merit but does not fully meet PLOS ONE’s publication criteria as it currently stands. Therefore, we invite you to submit a revised version of the manuscript that addresses the points raised during the review process.

Please address the concerns raised by two reviewers who are expert in this field. Please submit your revised manuscript by 24th April, 2022. If you will need more time than this to complete your revisions, please reply to this message or contact the journal office at plosone@plos.org. Please include the following items when submitting your revised manuscript:A rebuttal letter that responds to each point raised by the academic editor and reviewer(s). You should upload this letter as a separate file labeled 'Response to Reviewers'.A marked-up copy of your manuscript that highlights changes made to the original version. You should upload this as a separate file labeled 'Revised Manuscript with Track Changes'.An unmarked version of your revised paper without tracked changes. You should upload this as a separate file labeled 'Manuscript'.

We look forward to receiving your revised manuscript.

Kind regards,

Santosh Kumar

Academic Editor

PLOS ONE

“We are thankful to all the caregivers and children for participating in this study, data collectors for collecting the data, field supervisor for their tireless support, Zillur Rahman for administrative support, Mohammad Ashraful Islam for the managerial role. We would also like to acknowledge GAIN for technical support and BRAC for implementing the programme. icddr,b is grateful to the Governments of Bangladesh, Canada, Sweden, and the United Kingdom for providing core/unrestricted support. This research was funded by the Children’s Investment Fund Foundation (CIFF: UK). The views, opinions, assumptions, or any other information set out in this article are solely those of the authors and should not be attributed to CIFF or any persons connected with CIFF.”

“Include this sentence at the end of your statement: The funders had no role in study design, data collection and analysis, decision to publish, or preparation of the manuscript.”

6. PLOS requires an ORCID iD for the corresponding author in Editorial Manager on papers submitted after December 6th, 2016. Please ensure that you have an ORCID iD and that it is validated in Editorial Manager. To do this, go to ‘Update my Information’ (in the upper left-hand corner of the main menu), and click on the Fetch/Validate link next to the ORCID field. This will take you to the ORCID site and allow you to create a new iD or authenticate a pre-existing iD in Editorial Manager. Please see the following video for instructions on linking an ORCID iD to your Editorial Manager account: https://www.youtube.com/watch?v=_xcclfuvtxQ.

7. We note that [Figures 3-5] in your submission contain [map/satellite] images which may be copyrighted. All PLOS content is published under the Creative Commons Attribution License (CC BY 4.0), which means that the manuscript, images, and Supporting Information files will be freely available online, and any third party is permitted to access, download, copy, distribute, and use these materials in any way, even commercially, with proper attribution. For these reasons, we cannot publish previously copyrighted maps or satellite images created using proprietary data, such as Google software (Google Maps, Street View, and Earth). For more information, see our copyright guidelines: http://journals.plos.org/plosone/s/licenses-and-copyright.

a. You may seek permission from the original copyright holder of Figures 3-5] to publish the content specifically under the CC BY 4.0 license. 

Natural Earth (public domain): http://www.naturalearthdata.com.

8. Please include a caption for figure 5.

Reviewers' comments:

Reviewer's Responses to Questions

**Comments to the Author**

1. Is the manuscript technically sound, and do the data support the conclusions?

Reviewer #1: Partly

Reviewer #2: Partly

2. Has the statistical analysis been performed appropriately and rigorously? 

Reviewer #1: Yes

Reviewer #2: Yes

3. Have the authors made all data underlying the findings in their manuscript fully available?

Reviewer #1: Yes

Reviewer #2: Yes

4. Is the manuscript presented in an intelligible fashion and written in standard English?

Reviewer #1: Yes

Reviewer #2: Yes

5. Review Comments to the Author

Reviewer #1: While the results are interesting, the authors can better explain the contribution of this paper and clarify the details of the technical analysis, which I elaborate on along with additional comments in an attachment. I have also made references to further copyediting to improve the exposition of the paper.

Reviewer #2: This manuscript presents an interesting study investigating the geographic variations of food insecurity and some of the critical risk factors of food insecurity using a sample from 26 out of 64 districts of Bangladesh. The study is highlighting an important public health issue that is prevalent in many LMICs across the world. However, it remains unclear what gaps of knowledge in the current literature can be addressed by this study. Repetition of the same information (i.e., redundancy) is a critical problem found in this paper. The English needs to be improved substantially. Tables and figures need reorganization. In addition, several other areas require major revisions to make the paper acceptable for publication. The authors are recommended to consider the following comments and suggestions for making further revisions.

Abstract:

1) “We utilized pooled data……………………………..in 26 districts and two urban slums in Dhaka, Bangladesh.” This sentence is too long, please consider revisions. It is not necessary to introduce BRAC here. This has to be done in the introduction section. The authors can just say: “BRAC, one of the largest international NGOs located in Bangladesh….”.

2) “We used Household Food Insecurity Access Scale to estimate the food security status.” Please clarify the method further. Mention how surveys were conducted (face-to-face interview or self-reporting??).

3) “Hot spot analysis was conducted using the Getis-Ord Gi* statistic.” Clarify further for readers’ clear understanding. What was the purpose of the analyses and Getis-Ord Gi* statistic? Did the authors use these approaches to map the distribution of food insecurity in several geographically important regions in this country?

4) Line 39: Instead of “less chance” use “less likelihood”, a more statistically acceptable language.

5) Line 42-43: “poor population in the Northwestern, central and coastal parts of Bangladesh”. Should it be central or Central-Southwestern (i.e., Rajbari, Jessore, Faridpur etc.).

6) Line 43-45: Is the recommendation related to the findings of the study? I don’t think so. Write a recommendation that is consistent with your findings. The last sentence should be replaced with an appropriate statement.

Introduction:

1) The following two sentences do not provide any useful additional information in paragraph 1 (line 67-69). Consider rewriting them or delete these sentences. “The availability, accessibility, and utilization of foods…………………………………………….. the people in low- and middle-income countries (LMICs)”.

2) Paragraph 2 is excessively long and can be easily divided into two – the first one can talk about various health consequences of food insecurity and the next can justify why this study was so significant. Start a new paragraph from the following sentence: “As we see tremendous……….”.

3) In new paragraph 3, define districts for international readers in a sentence. Also, discuss food security disparity across the country as international readers may not be familiar with the geography of Bangladesh. Why do you think that food security is important for certain regions such as Northwestern, central, and coastal parts of Bangladesh?

4) Associations of food insecurity with diarrhea, respiratory symptoms and stunting were reported in a recent study in another part of South Asia, which are very relevant to Bangladeshi socioeconomic and cultural characteristics. Please include a sentence and the following reference: Chakraborty, Rishika & Weigel, M & Khan, Khalid. (2021). Food Insecurity Is Associated with Diarrhea, Respiratory Illness, and Stunting but Not Underweight or Obesity in Low-Resource New Delhi Households. Journal of Hunger & Environmental Nutrition. 1-16. 10.1080/19320248.2021.2008574.

5) The authors are addressing two issues here – risk factors of food insecurity and geographical variation of the same. Please separate the two problems clearly in Introduction and justify what risk factors were investigated for the Bangladeshi populations and high-risk geographical units.

6) BRAC should be introduced in the Introduction section and later in the paper only the abbreviated version (i.e., BRAC) should be used.

Methods:

1) Study design and setting: delete “’s (an international development organization) in Bangladesh”. As recommended, introduce BRAC in Introduction section.

2) Study design and setting: The first THREE sentences have overlapping information, redundancy is an issue here! Please re-write.

3) Study design and setting: “The evaluation method has been discussed in detail in other papers” – does not sound clear. What evaluation methods were the authors referring to? Please clarify.

4) Sample size and sampling section is too long. It can be easily shorted keeping only the key information. The section is also not very clear. Illustrate the sampling strategies first and then. Lots of overlapping information from the previous section, need major revision to make the narrative flow well.

5) The survey (how the interviews were conducted) procedure needs to be discussed in a separate section right after the sampling section.

6) Covariates measure; Consider eliminating the redundancy. For instance, “Additionally, we included the survey year variable (2015, 2016,169 2017, and 2018) in the multiple regression model to control time variations on food insecurity status since the surveys were conducted in different years”. This sentence should end right after “control time variations on food insecurity status” since the it is clear from the statement that the surveys were conducted across four years (mentioned in parentheses). In many other places the same problem (redundancy) is noticed and thorough revisions are therefore, recommended.

7) Data analysis: It is very important to describe the covariates that were included in the adjusted models. The authors described the model building procedures nicely but the covariates that were finally selected were not listed.

Results/Tables/Figures:

1) The caption of Figure 1 should be modified. It is not representing data from all 64 districts BUT from 26 districts. The suggested caption can be: “Prevalence of food insecurity status in 26 districts of Bangladesh that represented the study sample”.

2) Table can be shortened. For some variables it is recommended to include data for one category and the reader can assume the other. For example, household size ≥5 members can only be presented in the table, which automatically implies that the other category is <5 members. Please follow the same recommendation for other variables (except Wealth Index and Survey Year).

3) Figures 2 and 3 are representing more or less the same information. It is recommended that the authors should keep one of the two figures. If you keep figure 3, please indicate in the footnote that the white color code districts were not included in the study.

4) Table 2: Please indicate in footnote the names of the covariates that were included in the adjusted models. If the sets of covariates varied across the models include the list in the table in a new column or present them in a separate paragraph or supplementary table.

Discussion:

1) What exactly the current findings add to the literature? Was the study novel? Was this the first study in Bangladesh that identified the risk factors of food insecurity? If not, the authors need to justify the merit of the study in the first paragraph.

2) It is clear from the result section that the Northern districts are more vulnerable to food insecurity. Please provide rationale.

3) Strengths and limitations: The dataset was cross-sectional and temporality was not addressed due to only four years of data. Please address this limitation. Also, address lack of generalizability as a weakness since majority of the data were collected from rural populations.

4) The authors did not discuss the implications of the study from intervention design perspective. What interventions do you recommend based on the findings of the study?

5) The underlying risk factors of food insecurity are different for Northern and Southern districts/hot spots. The present study found more food insecurity in the Northern districts but highlighted the problems of the Southern/Coastal districts. Please discuss the critical environmental, geographic and community characteristics of the three distinct geographic regions – Northern, Southwestern and Costal regions (see figure 3 and 4). A major revision is needed in this area. Perhaps, the author might consider three separate paragraphs for the three regions mentioned above.

6) How are the risk factors identified by the authors explain the food insecurity for the above three vulnerable geographic regions? Please discuss.

6. PLOS authors have the option to publish the peer review history of their article (what does this mean?). If published, this will include your full peer review and any attached files.

Reviewer #1: No

Reviewer #2: No

---

## [Author Response · Author response to Decision Letter 0]

7 Aug 2022

Response to Academic Editor’s comments

Response: We have revised according to journal’s style.

Response: We have corrected in the revised submission.

“We are thankful to all the caregivers and children for participating in this study, data collectors for collecting the data, field supervisor for their tireless support, Zillur Rahman for administrative support, Mohammad Ashraful Islam for the managerial role. We would also like to acknowledge GAIN for technical support and BRAC for implementing the programme. icddr,b is grateful to the Governments of Bangladesh, Canada, Sweden, and the United Kingdom for providing core/unrestricted support. This research was funded by the Children’s Investment Fund Foundation (CIFF: UK). The views, opinions, assumptions, or any other information set out in this article are solely those of the authors and should not be attributed to CIFF or any persons connected with CIFF.”

“Include this sentence at the end of your statement: The funders had no role in study design, data collection and analysis, decision to publish, or preparation of the manuscript.”

Response: We have drooped the funding information from the acknowledgement section.

Response: We have updated the Data Availability statement in the revised submission.

Response: We have updated the Data Availability statement in the revised submission.

6. PLOS requires an ORCID iD for the corresponding author in Editorial Manager on papers submitted after December 6th, 2016. Please ensure that you have an ORCID iD and that it is validated in Editorial Manager. To do this, go to ‘Update my Information’ (in the upper left-hand corner of the main menu), and click on the Fetch/Validate link next to the ORCID field. This will take you to the ORCID site and allow you to create a new iD or authenticate a pre-existing iD in Editorial Manager. Please see the following video for instructions on linking an ORCID iD to your Editorial Manager account: https://www.youtube.com/watch?v=_xcclfuvtxQ.

Response: I have updated my ORCID ID.

7. We note that [Figures 3-5] in your submission contain [map/satellite] images which may be copyrighted. All PLOS content is published under the Creative Commons Attribution License (CC BY 4.0), which means that the manuscript, images, and Supporting Information files will be freely available online, and any third party is permitted to access, download, copy, distribute, and use these materials in any way, even commercially, with proper attribution. For these reasons, we cannot publish previously copyrighted maps or satellite images created using proprietary data, such as Google software (Google Maps, Street View, and Earth). For more information, see our copyright guidelines: http://journals.plos.org/plosone/s/licenses-and-copyright.

a. You may seek permission from the original copyright holder of Figures 3-5] to publish the content specifically under the CC BY 4.0 license. 

Natural Earth (public domain): http://www.naturalearthdata.com.

Response: Figure 3 is created using QGIS software and figures 4 and 5 were using ArcGIS software created by the author. These graphs were created using shapefile and don’t need any copy right issue.

8. Please include a caption for figure 5.

Response: We have added the caption in figure 5.

Reviewer #1: While the results are interesting, the authors can better explain the contribution of this paper and clarify the details of the technical analysis, which I elaborate on along with additional comments in an attachment. I have also made references to further copyediting to improve the exposition of the paper.

Response: Thank you so much for your valuable comments and suggestions. We have revised the manuscript by addressing your comments and according to your valuable suggestions. 

Summary and general evaluation

This paper aims to identify geographical variation in food insecurity at the district level and its correlates in Bangladesh. The authors utilize a GEE model to identify household-level factors associated with food security and hot spot analysis for spatial mapping of food insecurity. The main finding identifies the education of household heads or caregivers and poverty to be the key correlates of the food insecurity status of a household. While the results are interesting, the authors can better explain the contribution of this paper and clarify the details of the technical analysis, which I elaborate on below.

Response: Thank you so much for your valuable review and insightful comments and suggestions. We have tried to provide the contribution to the paper and clarified the technical analysis details in the revised version. 

Main comments

1) This paper’s contribution could be made more explicit. Towards the end of the introduction, the authors refer to existing studies on food insecurity in Bangladesh. The literature discussion would be more effective if used to contrast this paper’s relative contribution better. Hossain et al. (2020) also map the prevalence, gap, and severity of food insecurity in Bangladesh at the district level using survey and census data to find northern and southern parts more vulnerable. It would be useful to compare and illustrate the value addition of the methods used and findings of this paper (the indicator of food security is also different). For instance, the hot spot analysis identifies the central region as vulnerable to food insecurity in this paper. Is this due to the method used, year-specific changes (Hossain et al. (2020) use data from 2010 and 2011 whereas this paper uses data from 2015­­–2018) or the choice of outcome variable? Even if the reasons for the central region findings are hard to pin down, the value addition of this paper should be strengthened in the introduction.

Response: Thanks a lot for your valuable comments. We have revised the introduction section as follows:

A study presented the district-level food insecurity status using per capita calorie intake from the data of Household Income and Expenditure Survey 2010 and Bangladesh Population and Housing Census 2011 [19]. Although this study explored the district-level variations in the prevalence of food insecurity, it did not manifest the extent of food insecurity and associated correlates of the household's food insecurity status. Further, the estimates of food insecurity status were not based on the recent data. Our study presented food insecurity estimates from the recent data between 2015 to 2018 (page 5; lines: 99-105)

2) Measurement of food insecurity: There need to be more details on how the main outcome variable is constructed. It is not clear how the nine questions are used and what criterion is applied to determine the different levels of food insecurity. Are the hot spot analysis results sensitive to how the food insecurity criterion is applied? It would be useful also to discuss the advantages or limitations of what the outcome variable captures compared to alternate measures such as the per capita calorie intake used by the Bangladesh Bureau of Statistics (Hossain et al. 2020). If data is available, it would be interesting to see the robustness of results to alternative definitions of measuring food insecurity. 

Response: We already provided the nine questionnaires of the household hunger scale in the supplementary table (Supplementary Table 1) from where we constructed the food insecurity. However, according to your suggestion, we have elaborated on the outcome variable construction in the method section. These are as follows: -

We assessed the food insecurity status based on the last 30 days' recall responses from 9 questions (Supplementary Table S1). The questions were asked on how often the level of concern, lack of access to, variety, and quantity of food happened? The response to each question was scored ranging from 0 to 30. We made scoring of these responses as 0=0, 1-2=1, 3-10=2, and 11-30=3. The total score ranged from 0 to 27 for 9 questions. We then categorized the scores as 0-1=food secure, score 2/7=mild food insecure, 8/14=moderate food insecure, and score 15-27=severe food insecure households [26]. (page 8; lines: 166-172)

We believe that the Getis-Ord Gi* statistics is better suited for our dataset because both high and low values cluster spatially

We have also discussed the advantages and limitations of alternative measures i.e. food insecurity by using per capita calorie intake in the introduction section (page 5-6: Lines 106-110).

For your kind information, in our study, we don’t have the data for per capita calorie intake. Therefore, we missed the opportunity to see the robustness by using the alternative method as you suggested. 

3) It would be helpful to specify the main estimating equation in the methods section to clarify the control variables, including district and survey fixed effects. There can be additional discussion on why GEE models are used in this context.

Response: Thank you so much for your valuable suggestion. We applied the GEE model since we had clusters (PSU) data to capture the variations of clustering data. But we missed the opportunity to use weighted estimates in GEE.

In the revised version, we have changed our analysis of GEE to multivariable logistic regression since it will be a statistically rigorous method to adjust the clustering effect as well as weight the estimates. We performed this in the regression analysis by using “svy” prefix command in Stata. 

4) In the main GEE model, all covariates relevant in affecting food security (based on theory or previous literature) should be presented in the main model (regardless of statistical significance). Line 178 suggests that only statistically significant covariates from the simple GEE model were retained in the final multivariable model. 

Response: Thanks for your suggestion. We have revised our final regression model to include both statistically significant and conceptually or theoretically (even though not statistically significant) linked with the outcome. 

5) Is it possible to compare the magnitude of the estimates of food insecurity with other studies to contextualize the results? There is some reference to other estimates from the literature in lines 284¬–286. However, additional details are required to have more context regarding the time period and the comparable statistics with this study (if any). 

Response: We have revised the manuscript with the magnitude of the prevalence of other studies with our study in the following way:

“The prevalence of food insecurity status of these two studies were higher than our study since these studies were conducted during the COVID-19 period when the anxiety regarding food insecurity was higher among the people.” (pages 16-17; lines: 313-315)

6) In Figure 4, what does the blue area indicate? The map legend does not specify it, so it is hard to interpret.

Response: The blue area is due to clustering of households without food insecurity. The households without food insecurity was assigned a hallow blue circle in order to show distinct colors for two categories (solid red – households with food insecurity).

7) For the summary statistics in Table 1, it would be good to report the standard deviation of the variables.

Response: All the indicators in the summary statistics in Table 1 are categorical that’s why we presented them in percentages with frequencies. 

8) The exposition of the paper can be improved with some copy editing. I cite some select references, for e.g., the text in lines 71–73, 186–187, 197–199, 292 needs to be revised.

Response: Thanks a lot for pointing these. We have edited the selected lines in the revised version.

Minor comments

1) The abstract states that the food insecurity hotspots are in Bangladesh’s northwestern and coastal regions. However, when describing areas for targeted interventions, it refers to northwestern, central, and coastal regions. The main analysis indicates central regions also to be vulnerable to food insecurity. The abstract can be updated accordingly.

Response: Thanks for your valuable comment. According to your suggestion, we have included the Central-Southwestern region in the abstract. (page 2: line 45) 

2) In the abstract and line 113, it ought to be BRAC instead of BRAC’s.

Response: Thanks. We have made the change. (page 2: line:30) 

3) In line 138, it is noted that the method adopted is similar to the approach of the EPI. The way it is phrased at present, it is unclear what this refers to. 

Response: We agree with you that the method is almost similar to EPI method. We added the following sentence to make it clear

“We used this EPI-5 method to select households because we intended to see the coverage of a large evaluation programme”. (page 7; lines: 146-148)

4) Coastal spelled incorrectly in line 248.

Response: We are sorry for this error. Corrected. (page 13; line 268) 

5) Add detailed notes to the tables and figures to be self-contained, including information on the outcome variable.

Response: Thanks for your kind suggestion. We have added details notes in the tables and figures if necessary. 

6) Drop the open bracket in line 134 before PSUs.

Response: We have made the correction as you suggested. (page7; line 140) 

Reference:

Hossain, M. J., Das, S., Chandra, H., & Islam, M. A. (2020). Disaggregate level estimates and spatial mapping of food insecurity in Bangladesh by linking survey and census data. PloS One, 15(4), e0230906.

Reviewer #2: This manuscript presents an interesting study investigating the geographic variations of food insecurity and some of the critical risk factors of food insecurity using a sample from 26 out of 64 districts of Bangladesh. The study is highlighting an important public health issue that is prevalent in many LMICs across the world. However, it remains unclear what gaps of knowledge in the current literature can be addressed by this study. Repetition of the same information (i.e., redundancy) is a critical problem found in this paper. The English needs to be improved substantially. Tables and figures need reorganization. In addition, several other areas require major revisions to make the paper acceptable for publication. The authors are recommended to consider the following comments and suggestions for making further revisions.

Response: Thank you so much for your valuable comments and suggestions. We addressed your comments and made changes according to your suggestions in the revised version of this manuscript.

Abstract:

1) “We utilized pooled data……………………………..in 26 districts and two urban slums in Dhaka, Bangladesh.” This sentence is too long, please consider revisions. It is not necessary to introduce BRAC here. This has to be done in the introduction section. The authors can just say: “BRAC, one of the largest international NGOs located in Bangladesh….”.

Response: Thanks for your valuable comment. We have changed the sentence according to your suggestion. These are as follows:

“We utilized pooled data of seven cross-sectional surveys conducted at the household level from March 2015 to May 2018. This study was a part of the evaluation of the Maternal Infant Young Child Nutrition Phase 2 programme implemented by BRAC, one of the largest international NGOs located in Bangladesh that covered rural areas in 26 districts and two urban slums in Dhaka, Bangladesh.” (page 2; lines: 27-31).

2) “We used Household Food Insecurity Access Scale to estimate the food security status.” Please clarify the method further. Mention how surveys were conducted (face-to-face interview or self-reporting??).

Response: We have changed according to your suggestion. The changes are as follows¬¬¬:

We used Household Food Insecurity Access Scale (a widely used Scale to measure the household’s food insecurity) to estimate the food insecurity status from the information collected through a face-to-face interview by using a structured questionnaire. (page 2; lines: 31-34).

3) “Hot spot analysis was conducted using the Getis-Ord Gi* statistic.” Clarify further for readers’ clear understanding. What was the purpose of the analyses and Getis-Ord Gi* statistic? 

Response: We have elaborated the hot spot analysis in the main body of manuscript under data analysis section as follows:-

Hot spot analysis was conducted using the Getis-Ord Gi* statistic. The Getis-Ord Gi* statistics is better suited for our dataset because both high and low values cluster spatially [28]. Gi* statistics works by looking at each feature within the context (in our study all the study households) of neighboring features. A feature with a high value is interesting but may not be a statistically significant hot spot. To be a statistically significant hot spot, a feature will have a high value and be surrounded by other features with high values as well. The Gi* statistic is a z-score that identifies areas of higher or lower values by comparing them to a normal probability distribution and provides a measure of the local concentration of positive results. Each household location was assigned a value of “1” if the households fell in the insecure category or “0” if secured. For the conceptualization of spatial relationships, we used the Fixed Distance Band; this statistic compares spatial dependency of food insecurity between the households to identify hot spots and cold spots. A high z-score and small p-value for a feature indicate a spatial clustering of high values. A low negative z-score and small p-value indicate a spatial clustering of low values (cold spot). The higher (or lower) the z-score, the more intense the clustering (hot spot) [29]. The Getis-Ord Gi* statistic was used to classify households into hot spots and cold spots with 90%, 95%, and 99% confidence (page 10-11, lines: 210-225)

Did the authors use these approaches to map the distribution of food insecurity in several geographically important regions in this country?

Response: The analysis was done for Bangladesh as whole and not stratified by geographical regions.

4) Line 39: Instead of “less chance” use “less likelihood”, a more statistically acceptable language.

Response: According to your suggestion, we have replaced “less chance” with “less likelihood”. (page 2; line: 41).

5) Line 42-43: “poor population in the Northwestern, central and coastal parts of Bangladesh”. Should it be central or Central-Southwestern (i.e., Rajbari, Jessore, Faridpur etc.).

Response: Thanks. Specifically, it should be Central-Southwestern. We have changed central to Central-Southwestern in the revised version. (page 2; line: 45). 

6) Line 43-45: Is the recommendation related to the findings of the study? I don’t think so. Write a recommendation that is consistent with your findings. The last sentence should be replaced with an appropriate statement.

Response: Thanks for your comment. We have changed it accordingly as follows: -

Comprehensive interventions including the strategies for poverty reduction and education for all might be effective to reduce the food insecurity status at rural households in Bangladesh (page 3; lines: 46-47).

Introduction:

1) The following two sentences do not provide any useful additional information in paragraph 1 (line 67-69). Consider rewriting them or delete these sentences. “The availability, accessibility, and utilization of foods…………………………………………….. the people in low- and middle-income countries (LMICs)”.

Response: We deleted the first sentence and re-written the 2nd sentence as:

People who experience food insecurity are mostly from low- and middle-income countries (LMICs) [2]”. (page 4; lines: 68-69).

2) Paragraph 2 is excessively long and can be easily divided into two – the first one can talk about various health consequences of food insecurity and the next can justify why this study was so significant. Start a new paragraph from the following sentence: “As we see tremendous……….”.

Response: According to your valuable suggestion, we have split this paragraph into two. 

3) In new paragraph 3, define districts for international readers in a sentence. Also, discuss food security disparity across the country as international readers may not be familiar with the geography of Bangladesh. Why do you think that food security is important for certain regions such as Northwestern, central, and coastal parts of Bangladesh?

Response: We defined districts in the parenthesis. We discussed the rest of the part in the discussion section.

4) Associations of food insecurity with diarrhea, respiratory symptoms and stunting were reported in a recent study in another part of South Asia, which are very relevant to Bangladeshi socioeconomic and cultural characteristics. Please include a sentence and the following reference: Chakraborty, Rishika & Weigel, M & Khan, Khalid. (2021). Food Insecurity Is Associated with Diarrhea, Respiratory Illness, and Stunting but Not Underweight or Obesity in Low-Resource New Delhi Households. Journal of Hunger & Environmental Nutrition. 1-16. 10.1080/19320248.2021.2008574.

Response: Thanks for the kind suggestion. We have added the following sentence in the introduction section and cited your suggested article.

“A study has shown its association with diarrhea, respiratory illness, and stunting [13]”. (page 4; lines: 83-84).

5) The authors are addressing two issues here – risk factors of food insecurity and geographical variation of the same. Please separate the two problems clearly in Introduction and justify what risk factors were investigated for the Bangladeshi populations and high-risk geographical units.

Response: Thanks for your valuable suggestion. We have discussed the identified risk factors in para 3 in the introduction section. (page 5; lines: 87-93).

In the revised version in paragraph 4, we have discussed the studies conducted in Bangladesh, the findings, the variations of food insecurity and the knowledge gaps. (pages 5-6; lines: 95-118).

6) BRAC should be introduced in the Introduction section and later in the paper only the abbreviated version (i.e., BRAC) should be used.

Response: We have introduced BRAC in the Abstract and methods section and later used the term BRAC as you suggested.

Methods:

1) Study design and setting: delete “’s (an international development organization) in Bangladesh”. As recommended, introduce BRAC in Introduction section.

Response: We have deleted ’s (an international development organization) in Bangladesh from the revised manuscript.

2) Study design and setting: The first THREE sentences have overlapping information, redundancy is an issue here! Please re-write.

Response: Thanks for pointing out the redundancy issue. We have rewritten it in the following way: -

The study utilized pooled data collected at the household level, as part of the evaluation of the Maternal Infant and Young Child Nutrition (MIYCN) Phase 2 programme implemented by BRAC (an international development organization). The surveys were conducted from March 2015 to May 2018 in rural areas of 26 districts and two urban slums of Dhaka, Bangladesh. The surveys were conducted in three phases (baseline, midline, and endline) and, to limit the seasonal effect, surveys were conducted at the same time of the year as the previous surveys. The MIYCN programme used concurrent evaluation which is an innovative approrach to evaluate complex real-world programmes. The evaluation method including survey timelines, evaluation activities, evidence and course correction and study procedure has been discussed elsewhere [21,22]. (page 6; lines:122-130)

3) Study design and setting: “The evaluation method has been discussed in detail in other papers” – does not sound clear. What evaluation methods were the authors referring to? Please clarify.

Response: We have revised as: -

“The MIYCN programme used concurrent evaluation which is an innovative approrach to evaluate complex real-world programmes. The evaluation method including survey timelines, evaluation activities, evidence and course correction and study procedure has been discussed elsewhere [21,22]”. (page 6; line: 127-130)

4) Sample size and sampling section is too long. It can be easily shorted keeping only the key information. The section is also not very clear. Illustrate the sampling strategies first and then. Lots of overlapping information from the previous section, need major revision to make the narrative flow well.

Response: We separated the sample size and sampling section in the revised version. Also, we have shorten keeping with only key information in the sample size section. 

“Sample size 

We considered a 50% prevalence of MNP (micronutrient powder) coverage, a precision of ±10%, and a design effect of 2 for calculating the sample size. Our estimated minimum sample size was 192 households per district for caregivers of 6-59-month-old children. The detailed sample size calculation was presented in other papers of this project [23,24]. For this article, we used pooled data of seven surveys since the first survey of MIYCN did not include all the food insecurity indicators. In this study, we included 15,009 respondents as our study participants (page 7; lines: 132-137)

5) The survey (how the interviews were conducted) procedure needs to be discussed in a separate section right after the sampling section.

Response: Thanks for your kind suggestion. We included data collection section as follows: -

“Data collection

We interviewed the caregivers of the children aged 6-59 months. Before the interview, the interviewer read the aims, process of the study before the interview in the language interviewee understood. They were allowed to ask questions or raised any concerns about the study. After they were satisfied, interviewer took the written inform consent from the interviewee. We used structured questionnaire to collect the data. The questionnaire contains socio-demographic, health, and nutrition-related standard modules for caregivers, children, and households. The interview was administered by trained interviewers and lasted 40-60 minutes. To check the quality of data, the field research supervisor re-interviewed 6-10% of the original interviews. (pages 7-8; lines:151-158) 

6) Covariates measure; Consider eliminating the redundancy. For instance, “Additionally, we included the survey year variable (2015, 2016,169 2017, and 2018) in the multiple regression model to control time variations on food insecurity status since the surveys were conducted in different years”. This sentence should end right after “control time variations on food insecurity status” since the it is clear from the statement that the surveys were conducted across four years (mentioned in parentheses). In many other places the same problem (redundancy) is noticed and thorough revisions are therefore, recommended.

Response: Thanks a lot for your detailed comment. Deleted “since the surveys were conducted in different years”. Also, we tried to eliminate the redundancy in the revised version. (page 9; lines:190-192)

7) Data analysis: It is very important to describe the covariates that were included in the adjusted models. The authors described the model building procedures nicely but the covariates that were finally selected were not listed.

Response: According to your suggestion, we have listed the covariates in the following way-

“In the univariate models, household size, education of household heads, caregiver’s age, education of caregivers, wealth index, district, and survey year were significantly associated with food insecurity” (page 14; lines: 277-279)

“The covariates that were significantly associated with food insecurity in the simple regression model as well as other covariates that were not statistically significant but conceptually linked with households food insecurity, were included in the final multivariable regression model..” (page 10; lines: 200-203)

Results/Tables/Figures:

1) The caption of Figure 1 should be modified. It is not representing data from all 64 districts BUT from 26 districts. The suggested caption can be: “Prevalence of food insecurity status in 26 districts of Bangladesh that represented the study sample”.

Response: Changed the caption as “Prevalence of food insecurity status in 26 districts of Bangladesh”. Excluded the part “that represented the study sample” as you suggested since we talked sample size and sampling in the method section which reflects the repetitiveness of the sample. 

2) Table can be shortened. For some variables it is recommended to include data for one category and the reader can assume the other. For example, household size ≥5 members can only be presented in the table, which automatically implies that the other category is <5 members. Please follow the same recommendation for other variables (except Wealth Index and Survey Year).

Response: We have shortened Table 2 according to your suggestion. (page 13; lines:255-265)

3) Figures 2 and 3 are representing more or less the same information. It is recommended that the authors should keep one of the two figures. If you keep figure 3, please indicate in the footnote that the white color code districts were not included in the study.

Response: We agree with you. But from figure 2 we will get an actual estimate for severe, mild, and moderate food insecurity and food security prevalence. However, figure 3 provides, the prevalence with a given range and locations of districts from Bangladesh’s Map. Therefore, we think keeping both will help the readers as well as the policymakers to get actual estimates and vulnerable zones.

According to your suggestion, we indicate the while color districts were not included in our study. 

4) Table 2: Please indicate in footnote the names of the covariates that were included in the adjusted models. If the sets of covariates varied across the models include the list in the table in a new column or present them in a separate paragraph or supplementary table.

Response: According to your suggestion, we included the list of covariates that were included in the adjusted model in the footnote of Table 2.

Discussion:

1) What exactly the current findings add to the literature? Was the study novel? Was this the first study in Bangladesh that identified the risk factors of food insecurity? If not, the authors need to justify the merit of the study in the first paragraph.

Response: We dropped the first sentence as we have already mentioned the objective of this study in the last part of the introduction section.

According to your suggestion, we have included the novelty of this study in the first para as: -

“This is the first study that mapped the food insecurity status in rural districts of Bangladesh where the large-scale MIYCN programme has been implemented.” (page 16; lines: 295-296)

2) It is clear from the result section that the Northern districts are more vulnerable to food insecurity. Please provide rationale.

Response: We provided the rational as follows: - 

“The hot spot of food-insecure households was located in the Northwestern (Lalmonirhat, Nilphamari, Kurigram, Rangpur, Gaibandha, Mymensingh, and Sylhet), followed by Central-Southwestern (Rajbari, Faridpur, Magura, Jessore, and Madaripur) and coastal districts (Cox’s Bazar, Barguna) of Bangladesh. Most of the people in the regions have low-income (Lalmonirhat, Nilphamari, Kurigram, Rangpur, Gaibandha, and Mymensingh) and the literacy rate is also low (Sylhet) [35,36]. A majority portion of people living in the districts of Northern parts are suffering from a cyclical phenomenon of poverty and hunger called “Monga” [37]. Also, limited employment, high rate of landlessness, river erosions, floods, and droughts are the reasons for high food insecurity in this region.” (page: 17; lines: 316-325)

3) Strengths and limitations: The dataset was cross-sectional and temporality was not addressed due to only four years of data. Please address this limitation. Also, address lack of generalizability as a weakness since majority of the data were collected from rural populations.

Response: According to your suggestion, we have modified and included the limitations as follows:

“Data were collected mainly from the rural part of Bangladesh where a large scale MIYCN program was implemented and so, the results might not be representative for entire Bangladesh. Data collected from two urban slums in Dhaka that also do not necessarily represent the food insecurity status of urban areas in Bangladesh. To address these problems, further research is warranted to estimate urban food insecurity status. Furthermore, due to the cross-sectional nature and limited survey years, we cannot make any causal inferences as well as establish the temporality. We suggest a longitudinal study with a wider time frame to explore the cause-effect relationship and temporal trends of food insecurity as well as to monitor the food insecurity status and accordingly allocate the resources in the vulnerable districts in rural Bangladesh.” (pages: 19-20; lines: 369-377)

4) The authors did not discuss the implications of the study from intervention design perspective. What interventions do you recommend based on the findings of the study?

Response: We have provided the intervention strategies based on our identified factors as follows: -

“Therefore, interventions on poverty alleviation including cash and food transfer programme in the Northwestern, Central-Southwestern, and Coastal districts might be effective to reduce the food insecurity status [47]. Moreover, proper monitoring and strengthening of the school-based food for education programme might be effective to increase the literacy rate among the people of vulnerable areas of Bangladesh [48].” (page: 19; lines: 357-361) 

5) The underlying risk factors of food insecurity are different for Northern and Southern districts/hot spots. The present study found more food insecurity in the Northern districts but highlighted the problems of the Southern/Coastal districts. Please discuss the critical environmental, geographic and community characteristics of the three distinct geographic regions – Northern, Southwestern and Costal regions (see figure 3 and 4). A major revision is needed in this area. Perhaps, the author might consider three separate paragraphs for the three regions mentioned above.

Response: Thank you very much for your comments and suggestions.

In the revised version, we have included two paragraphs for describing the vulnerabilities of these three regions-Northern, and central-Southwestern and Costal regions. In the central-Southwestern region, the districts level information on geographical, environmental and community characteristics is rare. However, we search the literature and included few in a paragraph with Coastal districts. These are as follows: -

We found significant spatial variations in the prevalence of food insecurity status among 26 rural districts and two urban slums in Dhaka. The hot spot of food-insecure households was located in the Northwestern (Lalmonirhat, Nilphamari, Kurigram, Rangpur, Gaibandha, Mymensingh, and Sylhet), followed by Central-Southwestern (Rajbari, Faridpur, Magura, Jessore, and Madaripur) and coastal districts (Cox’s Bazar, Barguna) of Bangladesh. Most of the people in the regions have low-income (Lalmonirhat, Nilphamari, Kurigram, Rangpur, Gaibandha, and Mymensingh) and the literacy rate is also low (Sylhet) [35,36]. A majority portion of people living in the districts of Northern parts are suffering from a cyclical phenomenon of poverty and hunger called “Monga” [37]. Also, limited employment, high rate of landlessness, river erosions, floods, and droughts are the reasons for high food insecurity in this region. (page: 17; lines: 316-325)

“Most of the people in the coastal areas depend on fishing and farming agricultural lands for their livelihood. Frequent natural calamities and salinization of soil reduce crop production and pose long-term environmental degradation leading to food insecurity in these areas [39]. Following Northwestern and coastal districts, the food insecurity hotspot is also located in the Central-southwestern districts (Rajbari, Faridpur, Madaripur, Magura, Jessore). In these areas, floods, river bank erosions droughts are the consequences of climate change which might have effects on high food insecurity.” (page: 18; lines: 332-338) 

6) How are the risk factors identified by the authors explain the food insecurity for the above three vulnerable geographic regions? Please discuss.

Response: We have explained the identified risk factors for the three vulnerable areas as follows: -

“The education of household heads and poverty had greater consequences for reducing the food insecurity status of the vulnerable areas we identified. Because these areas are suffering from extreme poverty (Northwestern parts). Also, there is a direct link between poverty reduction and household head's education. Therefore, interventions on poverty alleviation including cash and food transfer programme in the Northwestern, Central-Southwestern, and Coastal districts might be effective to reduce the food insecurity status [47]. Moreover, proper monitoring and strengthening of the school-based food for education programme might be effective to increase the literacy rate among the people of vulnerable areas of Bangladesh [48].” (page: 19; lines: 353-361)

---

## [Decision Letter · Decision Letter 1]

24 Oct 2022

PONE-D-21-34476R1Geographical Variations of Food Insecurity and its Associated Factors in Bangladesh: Evidence from Pooled Data of Seven Cross-sectional SurveysPLOS ONE

Dear Dr. Tariqujjaman,

Thank you for submitting your manuscript to PLOS ONE. After careful consideration, we feel that it has merit but does not fully meet PLOS ONE’s publication criteria as it currently stands. Therefore, we invite you to submit a revised version of the manuscript that addresses the points raised during the review process.Referee 1 has raised additional concerns, and I would like you to address those concerns in the revised mansucript.

We look forward to receiving your revised manuscript.

Kind regards,

Santosh Kumar

Academic Editor

PLOS ONE

Journal Requirements:

Reviewers' comments:

Reviewer's Responses to Questions

**Comments to the Author**

1. If the authors have adequately addressed your comments raised in a previous round of review and you feel that this manuscript is now acceptable for publication, you may indicate that here to bypass the “Comments to the Author” section, enter your conflict of interest statement in the “Confidential to Editor” section, and submit your "Accept" recommendation.

Reviewer #1: (No Response)

2. Is the manuscript technically sound, and do the data support the conclusions?

Reviewer #1: (No Response)

3. Has the statistical analysis been performed appropriately and rigorously? 

Reviewer #1: (No Response)

4. Have the authors made all data underlying the findings in their manuscript fully available?

Reviewer #1: (No Response)

5. Is the manuscript presented in an intelligible fashion and written in standard English?

Reviewer #1: (No Response)

6. Review Comments to the Author

Reviewer #1: (No Response)

7. PLOS authors have the option to publish the peer review history of their article (what does this mean?). If published, this will include your full peer review and any attached files.

Reviewer #1: No

---

## [Author Response · Author response to Decision Letter 1]

8 Dec 2022

Summary and general evaluation

While the authors have addressed some of the comments, I note below where the authors still need to strengthen their arguments and incorporate additional changes to the manuscript.

Main comments

1) Contribution: While the authors have made some attempts to contrast their findings to that of Hossain et al. (2020), the paper’s contribution is not fully fleshed out yet and needs to be clearer and more explicit (beyond stating the use of more recent data). The writing needs to be re-framed so that, as a reader, it is clear what additional thing we learn from this paper relative to existing studies (whether its methodological innovation or otherwise). Other studies find similar results that food insecurity is prevalent in northern and southern parts of the country and that education and poverty are significant correlates of food insecurity, so the value addition needs to be clearer.

Response: Thank you for noting these. This study was conducted exclusively in the BRAC’s MIYCN programme areas and this study intended to understand the food insecurity in the areas despite having programmatic efforts in improving health outcomes. In addition, our study used Household Food Insecurity Access Scale whereas Hossain et. al. (2020) used per calorie intake method. We emphasized MIYCN programme areas in the revised version which we think will distinguish our study from others. 

We also reframed our manuscript, particularly in the discussion section for a clear understanding of the readers. 

2) Method and measurement: Please write down the main estimating equation in the methods section. While the control variables have been described much better now, the equation helps the reader to understand more clearly what you are capturing in your analysis. 

It’s good that there are additional details on the construction of the main outcome variable. However, lines 169 and 170 seem to provide conflicting numbers on whether the response to each question was scored from 0 to 30 or 0 to 27.

Response: Thank you for your valuable comments. In the revised manuscript, we have added the equation as follows—

The general equation of the multivariable logistic regression model is: 

〖y=log〗⁡[p/(1-p)]=α_0+β_1 x_1+ β_1 x_2+β_1 x_2+⋯+β_n x_n

Where: 

y = Outcome variable (food insecurity status)

p = probability of household to be food insure 

α_0 = Intercept (constant)

β_1, β_2, β_3,……β_n= Coefficients of the respective independent variables 

x_1,x_2, x_3,…x_n = Independent variables (Household size, household head’s age, education of household head…survey year, etc.) (page 10, lines 196-204)

Also, we have revised the scoring of the outcome variable as follows—

The questions were asked on how often the level of concern, lack of access to, variety, and quantity of food happened. The response to each question was between 0 to 30. We made scoring based on the responses as 0=0, 1-2=1, 3-10=2, and 11-30=3. Thus, the score ranged from 0 to 27 for 9 questions. We then categorized the scores as 0-1=food secure, score 2/7=mild food insecure, 8/14=moderate food insecure, and score 15-27=severe food insecure. (page 8, lines 162-166).

3) Exposition: Thank you for addressing some of the select copy editing notes I made last time. However, there is still significant scope for improving the exposition of the paper; for instance, there are still copy edits that have not been addressed [As an example, text in lines 209–210, lines 383–386]. There are some repetitive sentences in the manuscript. While I note some examples, the entire paper can benefit from further copy-editing.

Response: Thank you for your valuable suggestion. In the revised manuscript, we made substantial edits. In addition, we removed the repetitive sentences.

Minor comments

1) The tables have additional information now, which is helpful, but it is not yet fully self-contained. It should note what is being captured in the main outcome variable. A reader should be able to understand what is being estimated in the table based on the table notes. 

Response: Thanks a lot. We have added information in the footnote of Table 1 and additional information about the outcome variable in the footnote of Table 2.

---

## [Editor Report · Decision Letter 2]

21 Dec 2022

Geographical Variations of Food Insecurity and its Associated Factors in Bangladesh: Evidence from Pooled Data of Seven Cross-sectional Surveys

PONE-D-21-34476R2

Dear Dr Tariqijjaman, 

We’re pleased to inform you that your manuscript has been judged scientifically suitable for publication and will be formally accepted for publication once it meets all outstanding technical requirements.

Kind regards,

Santosh Kumar

Academic Editor

PLOS ONE
---

## [Editor Report · Acceptance letter]

28 Dec 2022

PONE-D-21-34476R2 

Geographical Variations of Food Insecurity and its Associated Factors in Bangladesh: Evidence from Pooled Data of Seven Cross-sectional Surveys 

Dear Dr. Tariqujjaman:

I'm pleased to inform you that your manuscript has been deemed suitable for publication in PLOS ONE. Congratulations! Your manuscript is now with our production department. 

Kind regards, 

on behalf of

Dr. Santosh Kumar 

Academic Editor

PLOS ONE